# Italians Are the Fastest 3000 m Open-Water Master Swimmers in the World

**DOI:** 10.3390/ijerph18147606

**Published:** 2021-07-17

**Authors:** Aldo Seffrin, Claudio A. B. Lira, Rodrigo L. Vancini, Douglas A. T. Santos, Cathia Moser, Elias Villiger, Thomas Rosemann, Beat Knechtle, Lee Hill, Marilia S. Andrade

**Affiliations:** 1Department of Physiology, Federal University of São Paulo, São Paulo 04021-001, Brazil; netoseffrin@gmail.com (A.S.); marilia1707@gmail.com (M.S.A.); 2Human and Exercise Physiology Division, Faculty of Physical Education and Dance, Federal University of Goiás, Goiânia 74690-900, Brazil; andre.claudio@gmail.com; 3Center for Physical Education and Sports, Federal University of Espírito Santo, Vitória 29075-910, Brazil; rodrigoluizvancini@gmail.com; 4Faculty of Physical Education, State University of Bahia, Teixeira de Freitas 45995-000, Brazil; datsantos@uneb.br; 5Balgrist University Hospital, 8008 Zurich, Switzerland; cathiamoser@gmx.ch; 6Institute of Primary Care, University Hospital Zurich, 8091 Zurich, Switzerland; evilliger@gmail.com (E.V.); thomas.rosemann@usz.ch (T.R.); 7Medbase St. Gallen Am Vadianplatz, 9000 St. Gallen, Switzerland; 8Division of Gastroenterology and Nutrition, Department of Pediatrics, McMaster University, Hamilton, ON L8S 4L8, Canada; hilll14@mcmaster.ca

**Keywords:** long-distance swimming, nationality, performance, age group athlete

## Abstract

Background: It is well known that athletes originating from a specific region or country can master specific sports disciplines (e.g., East-African runners in long-distance running). In addition, physical and athletic performance are the result of an interaction between genetic, environmental and epigenetic factors. However, little is known about on what determines sports success and performance for long-distance master swimmers such as origin. The aim of the study was to investigate the participation and performance trends of elite master open-water swimmers competing in the World Championships (WC) in 3000 m open-water swimming between 1986 and 2019. Methods: A total of 9247 valid participants were analyzed using generalized linear models (GLMs) with a gamma probability distribution and log link function. Resultsː Most of the starters were from Italy (1646 participations), followed by the United States of America (USA) (1128 participations) and Germany (959 participations). Swimmers from Italy were significantly faster than swimmers from Canada, Germany, USA, Great Britain and also from all other countries grouped (*p* < 0.005). The age group from 35–39 years old was significantly faster than athletes from age groups of 25–29 years old, 30–34 years old, 40–44 years old, 45–49 years old and 50–54 years old (*p* < 0.005). The percentage of local athletes in WC was 36% and varied from 36% (Italy, 2004) to 43 % (Germany), 53% (Italy, 2012) and up to 68 % (USA, 1992). Conclusions: Swimmers from Italy were the faster and the most numerous starters during this period of 27 years and 15 editions all over the world in 3000 m master open-water swimming.

## 1. Introduction

It is an interesting phenomenon that athletes originating from a specific region or country are dominating certain sports disciplines [1]. In running, for example, athletes originating from Jamaica have historically dominated in sprint events (100 m, 200 m and 400 m) [2], whereas athletes from East Africa (i.e., Ethiopia and Kenya) are some of the fastest runners over middle distance [3] and the marathon distance [4,5]. Interestingly, runners from Russia have been recently shown to consistently outperform ultra-runners from other regions in 100 km ultra-marathons [6] and the Comrades Marathon [7].

Furthermore, it seems that some of these specific athletes are from very distinct regions within a given country. For example, within Jamaica, sprint athletes predominantly originated from the Surrey County, whilst middle distance athletes were found to be over-represented from the Middlesex County [2]. Similarly, among East African runners, most national and international Kenyan athletes came from the Rift Valley province [3,8], and Ethiopian marathoners originated mainly from the regions of Arsi and Shewa [9]. 

However, for swimming, very little is known regarding the relationship between the origin of an athlete and its performance; moreover, most of the studies are from pool swimmers’ performance [10,11,12,13,14], and little is known about open water swimmers. In one of the few previous studies about open water swimmers, it was demonstrated that most of the athletes who participated in the English Channel Crossing between 1875 and 2013, were from Great Britain, the United States of America (USA), Australia and Ireland [15], and the fastest swim times were achieved by athletes from the USA, Australia and Great Britain [15]. In the Strait of Gibraltar, local Spanish swimmers were recorded as the fastest to complete the crossing [16]. Despite this, these open water swimming events garner interest from international participants, and many local swimmers participated and were among the fastest finishers. In this direction, it would be very interesting to know the performance of swimmers in an event that takes place worldwide, such as World Championships (WC). The knowledge about the origin of the most successful athletes allows future studies to be developed in order to evaluate the specificities of these regions/peoples capable of generating differentiated results in aquatic marathon events. In addition to performance, it is important to take in mind that master open water events also have the important aim to promote fitness among swimmers above 25 years old, therefore, it is also especially important to know which countries have the largest number of athletes participating in master swimming events. 

For master swimmers, it is well-known that their performance declines with increasing age [17,18]. This age-related performance decline with advancing age is due to a decline in physiological functional capacity [17]. However, this decline might be slowed down with increased training volume [19]. For 3000 m open-water swimmers, the performance trend by age group has already been investigated [20] and we know the sex difference in open-water swimming [21], however, we have no knowledge in which age group these 3000 open-water master swimmers achieve their best performance.

Therefore, the aim of the study was to investigate the participation and performance trends of master open-water swimmers competing in the WC in 3000 m open-water swimming. The study also aimed to compare the performance of the swimmers among different age groups. Based upon existing findings for pool and open-water swimmers, we hypothesized that the fastest swimmers competing in this discipline would originate from the USA and Australia.

## 2. Materials and Methods

### 2.1. Ethical Approval

The study was conducted according to the guidelines of the Declaration of Helsinki. This study was approved by the Institutional Review Board of Kanton St. Gallen, Switzerland, with a waiver of the requirement for informed consent of the participant as the study involved the analysis of publicly available data (EKSG 01-06-2010).

### 2.2. Data

Race results from all female and male master swimmers competing in the 3000 m open-water events from the Fédération Internationale De Natation (FINA) World Master Championships (WC), held between 1992 and 2019, were obtained from the official website [22]. 

Table 1 shows the year of the event and the location. Although the first FINA WC was held in 1986 in Tokyo, Japan, the first 3000 m open-water events were held in 1992 in Indianapolis, USA. A total of 9687 results were analyzed. The final race time of 200 participants, age of 239 participants and the nationality of one participant was not available. Thus, the final data set resulted in 9247 valid participants.

### 2.3. Statistical Analysis

The primary analytical approach was a descriptive analysis. Data were described in absolute and relative frequency and means with confidence intervals (95%). Then, generalized linear models (GLMs) with a gamma probability distribution and log link function were used to assess the effect of age groups and the effect of the athlete’s origin on swimming time, sex and whether they were a local part of the models. Differences found were investigated with the Post-hoc Bonferroni test. Possible interactions with sex or the fact that the participants were local were also investigated in each analysis. The Akaike Information Criterion (AIC) was used to choose the distribution of the dependent variable and the model linking function, using its lowest value [23]. In addition, Omnibus test was used to ensure that the model outperforms the null model. *p* < 0.05 was considered significant, and all *p*-values were 2-sided. Statistical analyses were performed using IBM SPSS Statistics (version 26, IBM SPSS, Chicago, IL, USA).

## 3. Results

The ten best times among elite male master open-water swimmers competing in the WC in 3000 m open-water swimming between 1992 and 2019 were from five different countries. They were Italy (five athletes), Germany (two athletes), Russia (one athlete), Switzerland (one athlete) and Austria (one athlete). The ten best times for female were from seven different countries. They were from Italy (three athletes), Great Britain (two athletes), Switzerland (one athlete), the Netherlands (one athlete), Germany (one athlete), USA (one athlete) and Austria (one athlete). Curiously, all the male and female ten best times in 3000 m open-water swimming were achieved in the same WC, which was held in Riccione, Italy, 2012. 

Comparing the 10 best times from each country, Italy showed significantly faster race times than athletes from Canada, Germany, USA, Great Britain, and all other countries (Figure 1).

The number of participants from each country in each year varied between 1 and 1023. The three countries that most participated in 3000 m open-water swimming WC over the years where Italy had 1646 participations, the USA had 1128 participations and Germany had 959 participations (Table 2).

The age group that showed the best times in relation to several other age groups was the group of individuals from 35 to 39 years old (Figure 2).

Analyzing local and foreign athletes’ participation, we observed that there was an average participation of 36% of local athletes in the WC. When the WC was held in the USA, the local athletes’ participation was 68% (1992) and 57% (2006); in Germany 43%, and in Italy 36% in 2004 and 53% in 2012 (Table 3).

An interesting piece of complementary information was that countries that had never participated in other events participated when they were hosts (South Korea and Morocco). The increase in the number of local athletes participating in the WC also coincides with the increase in the number of athletes that figure in the top ten for USA, Great Britain, Italy, Germany, Austria, Canada and Russia but not for Hungary, Morocco, New Zealand, Sweden and Korea.

## 4. Discussion

The aim of the study was to investigate the participation and performance trends of master open-water swimmers competing in the WC in 3000 m open-water swimming with the hypothesis that the fastest swimmers competing in that discipline would originate from the USA and Australia.

The main findings were (i) swimmers from Italy were significantly faster than swimmers from Canada, Germany, USA, Great Britain and also from all other countries grouped; (ii) age group from 34–39 years old was faster than the younger age groups; (iii) most of the starters were from Italy (1646 participations), followed by the USA (1128 participations) and Germany (959 participations); (iv) the ten fastest female and male race times were achieved in the same WC, in Riccione, Italy, in 2012; (v) the ten fastest female race times were from Italy, Great Britain, Switzerland, Netherlands, Germany, USA and Austria; (vi) the ten fastest male race times were from Italy, Germany, Russia, Switzerland and Austria, and (vii) the percentage of local athletes in local WC was, on average, 36%.

The most important finding was that swimmers from Italy were significantly faster than swimmers from other nationalities and they were also the most numerous starters during this period of 27 years and 15 editions worldwide. This contrasts with our hypotheses that the fastest swimmers in this discipline would originate from the USA and Australia [10,11]. A first explanation based upon our findings could be that most of the participants were originating from Italy, although it hosted the WC only two times during these 15 editions. There is also the question of the ease of access to competitions on the European continent. Europe has this advantage of the congregation of nations, many of them elite in sports. In other words, it is one thing to live in Europe and go to a competition in Europe, it is another to leave Europe and go to the USA and/or Australia [24,25].

A potential explanation that Italian swimmers were the fastest in the same Championship (Riccione, 2012) could be that climate and environmental conditions (e.g., water temperature) were quite favorable in that race [26,27]. Open-water swimming races are influenced by extreme environmental conditions (e.g., water temperature, tides, currents and waves) with an impact on performance, tactics and pacing [27]. Time or velocity comparison can be misleading in open-water swimming events due to the changing external conditions (e.g., water temperature, currents, circuit structure, etc.) between races. Most probably, external conditions were outstanding in that specific race and local athletes could profit. However, this does not explain why swimmers from other countries were not better than the Italian swimmers since environmental conditions were the same for all swimmers in that specific race.

Regarding swimming performance, Italy takes the 9th position in the world with 33 titles in competitive swimming over the course of its participation in international-level swimming events and the Summer Olympics [28]. Considering the world’s leading nation in swimming, Italy has improved from 9th position in 2017 to 4th position in 2019 [29]. However, future studies might investigate the participation and performance trends in elite Italian swimmers competing at world-class level in WC and Olympic Games for elite swimmers.

However, the Italians were the fastest swimmers and the most numerous in this specific sports discipline. It might be argued that Italy has a specific program for swimmers or even master swimmers [30]. The Italian Swimming Federation—Federazione Italiana Nuoto (FIN) has a specific section for master swimmers [31] with Italian Championships for master swimmers [32]. Furthermore, Riccione has, since 2003, a master team of the Polisportiva Riccione which was born from the passions of a group of swimmers. Master Nuoto Riccione organizes different races and prepares master swimmers for national and international events [33].

It is well-known that exercise and physical activity are associated with better quality of life and health outcomes [34]. There is also the question where countries have a transition program for “retirement from sport”, that is, prepare an elite athlete to be a “normal person” from the social and biological point of view [35,36]. Preparation programs for “retirement from sport” are essential for at least one athlete to continue to perform a minimum of physical activity for the maintenance of physical and mental health. Perhaps Italy is a model.

Apart from Italy, master swimming is also supported in other countries. In the USA, U.S. Masters Swimming promotes open-water master swimming providing details of clubs, events and training for future open-water master swimmers [37]. In England, master swimmers can find a club and information about swimming technique and nutrition [38]. Most probably, there are also, in other countries besides Italy, very specific regions where open-water master swimmers are specifically supported. In Oregon, USA, Oregon Masters Swimming promotes lifelong fitness by offering organized swimming for adults of all ages [39]. Additionally, in Wales, master swimmers are supported specifically with information about swimming techniques and events [40]. Eventually, these organizations also provide specific information about nutrition in open-water swimming [41]. 

Swimming is an interesting activity, and is deeply rooted in cultural practices and bodily encounters that change in meaning and over time and from place to place [42]. Over the course of history, humans form relationships with bodies of water (pools, lakes, rivers, oceans) and learn to be a body within that water [42]. Within that context, we understand the importance of bodies of water in terms of both their physical and mental benefits [43,44]. Although many long-distance open-water swimming events are solo feats, swimming is a social sport, situated in meaningful cultural practices [45]. Many countries located along coastlines or near bodies of water have a local population that utilizes these bodies for leisure and physical activity [46]. It is possible that countries who routinely perform the best during long-distance open-water swimming events are those whose cultural practices are situated within these bodies of water or, at the very least, originate from countries with strong sport and physical activity policies [47]. This could, in part, explain some of the findings in this study. Similar to many other countries, Italy is located along the warm waters of the Mediterranean Sea, where swimming, fishing, and bathing have strong cultural importance [48,49]. As a result, this could produce the perfect environment for Italian swimmers to refine their swimming abilities and contribute to world-class performances in international events. 

Finally, another important finding from the present study is that the 3000 m WC does not only have the participation of local athletes but, on average, the participation of locals in the events was 36%; that is, the majority of participants in each event were foreigners. This result is very interesting because it shows that these events really encourage the participation of master athletes from around the world, contributing to promotion of fitness among swimmers above 25 years old. This incentive for the sport of master athletes is of great social importance, given the known benefits of physical activity throughout the aging process. The success of creating these master competition events and the goal of promoting physical conditioning among older individuals is also evidenced when the results show that the 35–39-year-old athlete group were faster than the younger athlete groups (25–29 years and 30–34 years).

A limitation of the study is the fact that data from the first FINA World Master Championship held in 1986 in Tokyo, Japan, were not included. Future studies need to investigate the participation and performance trends in elite Italian swimmers competing at world-class level in WC and Olympic Games. A further limitation is the fact that environmental conditions such as water temperature, tides, currents and waves have not been considered. All these conditions might have an overall impact on performance, influencing tactics and pacing [27]. The findings for master swimmers competing in the FINA World Master Championships do not allow drawing conclusions for elite athletes competing in World or Continental Championships.

## 5. Conclusions

In the WC in 3000 m open-water swimming between 1986 and 2019, swimmers from Italy achieved the ten fastest race times in the same WC. Athletes from Italy were the most numerous starters during this period of 27 years and 15 editions all over the world in 3000 m master open-water swimming. Athletes in the age group 35–39 years were the fastest and faster than the younger age groups of 25–29 years and 30–34 years.

## Figures and Tables

**Figure 1 ijerph-18-07606-f001:**
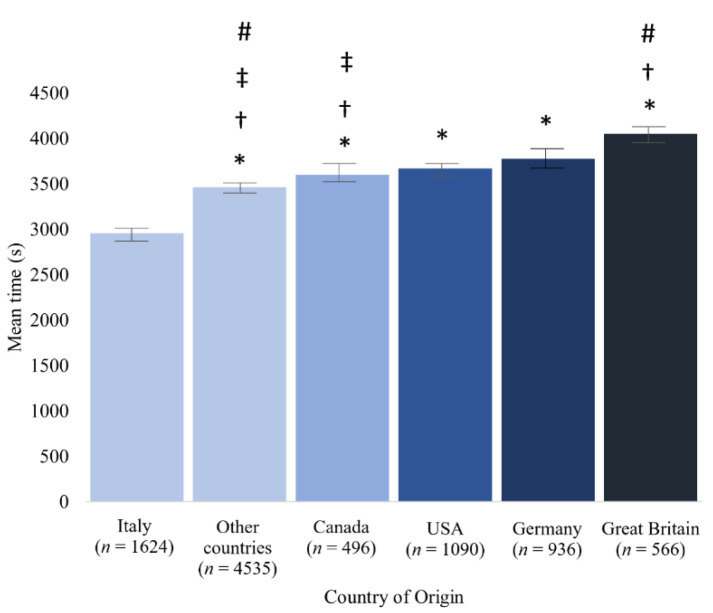
Mean times, confidence interval (95%) and difference between countries in 3000 m open-water swimming between 1992 and 2019. * different from Italy; † different from Germany; ‡ different from Great Britain; # different from USA.

**Figure 2 ijerph-18-07606-f002:**
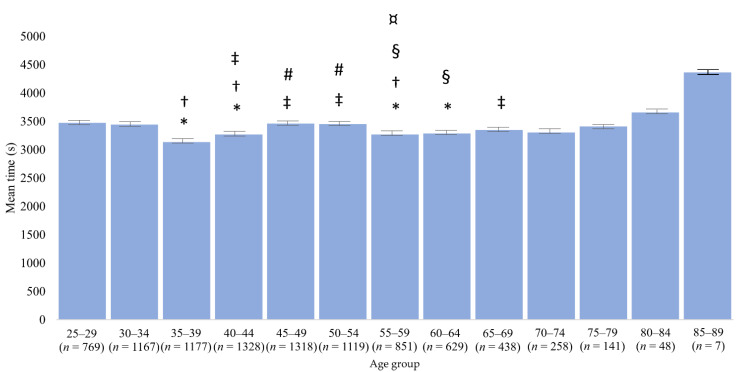
Mean times, confidence interval (95%) and difference between age groups in 3000 m open-water swimming WC between 1992 and 2019. * different from 25–29; † different from 30–34; ‡ different from 35–39; # different from 40–44; § different from 45–49; ¤ different from 50–54.

**Table 1 ijerph-18-07606-t001:** Location of the FINA World Master Championships 1992–2019.

Year	Time of Year	Location (City and Country)	Season
1992	25 June–5 July	Indianapolis (United States of America, USA)	Summer
1994	4–10 July	Montreal (Canada, CAN)	Summer
1996	23 June–3 July	Sheffield (Great Britain, GBR)	Summer
1998	19–30 June	Casablanca (Morocco, MAR)	Summer
2000	29 July–4 August	Munich (Germany, GER)	Summer
2002	21 March–3 April	Christchurch (New Zealand, NZL)	Autumn
2004	1–13 June	Riccione (Italy, ITA)	Spring
2006	4–17 August	Stanford (USA)	Summer
2008	18–25 April	Perth (Australia, AUS)	Autumn
2010	27 July–7 August	Gothenburg–Borås (Sweden, SWE)	Summer
2012	3–17 June	Riccione (Italy, ITA)	Spring
2014	27 July–10 August	Montreal (Canada, CAN)	Summer
2015	5–16 August	Kazan (Russia, RUS)	Summer
2017	7–20 August	Budapest (Hungary, HUN)	Summer
2019	5–18 August	Gwangju (South Korea, KOR)	Summer

**Table 2 ijerph-18-07606-t002:** Number of athletes from each country who participated in the WC each year (data are from the ten countries that most participated).

Race	AUS	BRA	CAN	ESP	FRA	GBR	GER	ITA	RUS	USA	Total
1992—USA	14	4	13	1	2	3	6	2	1	195	241
1996—GBR	12	21	12		10	97	27	8	1	18	206
1998—MAR	19	8		8	11	21	32	8	5	16	128
2000—GER	5	10	6	8	18	29	186	33	1	31	327
2002—NZL	57	8	9	3	7	19	21	6	1	24	155
2004—ITA	36	16	8	23	52	52	124	323	17	35	686
2006—USA	18	14	24	11	20	33	32	10	4	407	573
2008—AUS	272	3	15	7	36	38	31	7	7	56	472
2010—SWE	18	19	19	46	46	66	138	63	23	42	480
2012—ITA	32	34	20	52	84	75	131	1023	42	48	1541
2014—CAN	19	33	343	12	98	37	67	32	6	183	830
2015—RUS	12	5	2	14	3	7	36	15	171	8	273
2017—HUN	22	50	26	74	67	91	95	105	103	44	677
2019—KOR	11	15	9	23	15	26	33	11	23	21	187
Total	547	240	506	282	469	594	959	1646	405	1128	6776

AUS—Australia; BRA—Brazil; CAN—Canada; ESP—Spain; FRA—France; GBR—Great Britain; GER—Germany; HUN—Hungary; ITA—Italy; KOR—South Korea; MAR—Morocco; NZL—New Zealand; RUS—Russia; SWE—Sweden.

**Table 3 ijerph-18-07606-t003:** Total number of participating athletes and local athletes in each one of the 3000 m open- water WC.

Race	Participants	Locals	% of Locals
1992—USA	287	195	68%
1996—GBR	283	97	34%
1998—MAR	198	5	3%
2000—GER	430	186	43%
2002—NZL	307	97	32%
2004—ITA	899	323	36%
2006—USA	709	407	57%
2008—AUS	657	272	41%
2010—SWE	813	61	8%
2012—ITA	1944	1023	53%
2014—CAN	1072	343	32%
2015—RUS	367	171	47%
2017—HUN	1214	139	11%
2019—KOR	507	178	35%
Mean			36%

AUS—Australia; CAN—Canada; GBR—Great Britain; GER—Germany; HUN—Hungary; ITA—Italy; KOR—South Korea; MAR—Morocco; NZL—New Zealand; RUS—Russia; SWE—Sweden; USA—United States of America.

## Data Availability

Race results from all female and male master swimmers competing in the 3000 m open-water events from the Fédération Internationale De Natation (FINA) World Master Championships, held between 1992 and 2019, were obtained from the official website (www.fina.org/content/fina-masters-world-championships-results-archive; accessed on 1 March 2021).

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
