# Peer review of "Italians Are the Fastest 3000 m Open-Water Master Swimmers in the World"

_ijerph, 2021, doi:10.3390/ijerph18147606_

Round 1

Reviewer 1 Report

General comments

I do not have any further particular concerns to express about the manuscript. Authors addressed sufficiently the two minor points raised by me.

Author Response

Reviewer 1

General comments

I do not have any further particular concerns to express about the manuscript. Authors addressed sufficiently the two minor points raised by me.

Answer: We thank the expert reviewer for his/her comments, no further changes are required.

Reviewer 2 Report

Changes made by authors in the present manuscript do not solve the important flaws in the research desing and the rationale of the study. Authors responded to the reviewer comments referring to the "reviewer personal opinion" which is completely unacceptable in relation to: 

  • Methodological mistake of comparing average velocity between different open water races where environmental conditions may change (Baldassarre, 2017)
  • Making parallelism between elite athletes and master athletes to justify the present research, considering Master Swimming is a discipline developed by FINA (Federation Internationale Natation) to promote fitness among swimmers above 25 years old.
  • Drawing conclusions from the nationality of swimmers in a non-official event that is not celebrated in any official competition like World or Continental Championships. 

Author Response

Reviewer 2

Changes made by authors in the present manuscript do not solve the important flaws in the research desing and the rationale of the study. Authors responded to the reviewer comments referring to the "reviewer personal opinion" which is completely unacceptable in relation to: 

  • Methodological mistake of comparing average velocity between different open water races where environmental conditions may change (Baldassarre, 2017)
  • Making parallelism between elite athletes and master athletes to justify the present research, considering Master Swimming is a discipline developed by FINA (Federation Internationale Natation) to promote fitness among swimmers above 25 years old.
  • Drawing conclusions from the nationality of swimmers in a non-official event that is not celebrated in any official competition like World or Continental Championships. 

Answer: We agree with the expert reviewer and add this last aspect in the limitations.

Reviewer 3 Report

This version significantly improved compared to the previous one, congratulations. I believe it is close to publication, however, some details which require correction or improvement continue to be observed. Most are details, but the improvement of introduction section and principally the error in numbering of references is very relevant.

Line 60 - “USA” the first-time appearance in text should be in full.

Line 57 - Ref 11 / Line 60 - Ref 15. This is not according to journal guidelines. Please carefully reformulate all references.

Line 63 - Ref 16 / Line 106 - Ref 19. This is not according to journal guidelines. Please carefully reformulate all references.

Since the study “aimed to compare the performance of the swimmers among different age groups” Please consider improving the introduction section with text related to aging and physical capacities in swimming.

Line 96 - Please consider USA in full, standardized with other countries in the table.

Line 66 - Please consider separate all the main findings with “,” or “;”.

Line - 140 - The letter size is too small, can you please consider the same size as in figure 1?

Line 244 - WC (abbreviation)

Line 268 - “V..;” (only 1 end point). Otherwise, “H” does not have end point after.

Line 339 - Please carefully double check the references in the final. One example, Journal of Applied Sport Psychology has abbreviation (J. Appl. Sport Psychol).

Please, after improving the manuscript, consider a careful read with emphasis in English improvement.

Author Response

Reviewer 3

This version significantly improved compared to the previous one, congratulations. I believe it is close to publication, however, some details which require correction or improvement continue to be observed. Most are details, but the improvement of introduction section and principally the error in numbering of references is very relevant.

Line 60 - “USA” the first-time appearance in text should be in full.

Answer: We agree with the expert reviewer and changed as requested

Line 57 - Ref 11 / Line 60 - Ref 15. This is not according to journal guidelines. Please carefully reformulate all references.

Answer: We agree with the expert reviewer and changed as requested

Line 63 - Ref 16 / Line 106 - Ref 19. This is not according to journal guidelines. Please carefully reformulate all references.

Answer: We agree with the expert reviewer and changed as requested

Since the study “aimed to compare the performance of the swimmers among different age groups” Please consider improving the introduction section with text related to aging and physical capacities in swimming.

Answer: We agree with the expert reviewer and added a section in the introduction and comment these findings also in the conclusions.

Line 96 - Please consider USA in full, standardized with other countries in the table.

Answer: We agree with the expert reviewer and changed as requested

Line 66 - Please consider separate all the main findings with “,” or “;”.

Answer: We agree with the expert reviewer and changed as requested

Line - 140 - The letter size is too small, can you please consider the same size as in figure 1?

Answer: We agree with the expert reviewer and changed as requested

Line 244 - WC (abbreviation)

Answer: We agree with the expert reviewer and changed as requested

Line 268 - “V..;” (only 1 end point). Otherwise, “H” does not have end point after.

Answer: We agree with the expert reviewer and changed as requested

Line 339 - Please carefully double check the references in the final. One example, Journal of Applied Sport Psychology has abbreviation (J. Appl. Sport Psychol).

Answer: We checked all references again

 Please, after improving the manuscript, consider a careful read with emphasis in English improvement.

Answer: We checked the manuscript again

Round 2

Reviewer 3 Report

Congratulations for the work developed by the authors and editor regarding this manuscript.

This manuscript is a resubmission of an earlier submission. The following is a list of the peer review reports and author responses from that submission.

Round 1

Reviewer 1 Report

This is an interesting paper about some descriptive data, that should be interesting to present in a conference. However, I am afraid that the results are not unknown and this is a pure descriptive study and it lacks further investigation/discussion.  I think the manuscript, in the way that is presented, do not met the needed quality to be published in the current Journal. 

Author Response

Reviewer 1

This is an interesting paper about some descriptive data, that should be interesting to present in a conference. However, I am afraid that the results are not unknown and this is a pure descriptive study and it lacks further investigation/discussion.  I think the manuscript, in the way that is presented, do not met the needed quality to be published in the current Journal. 

Answer: We disagree with the expert reviewer. A search in GOOGLE showed no similar results. This is the first study to investigate this topic.

Reviewer 2 Report

General comments

This manuscript aims at investigating the participation and performance trends of elite master open-water swimmers competing in the World Championships in 3000 m open-water swimming. It is worthwhile to highlight the valuable attempt to provide a humanistic explanation of why “Italians are the fastest 3000 m open-water master swimmers in the world”. Overall, authors manage to fulfill sufficiently their aim.

The authors could try to provide some further insights regarding the diffusion of masters sports in countries most successful in master open-water swimming. One further suggestion could be to try to find information regarding nutrition habits in master athletes in countries most successful in master open-water swimming.

Author Response

Reviewer 2

General comments

This manuscript aims at investigating the participation and performance trends of elite master open-water swimmers competing in the World Championships in 3000 m open-water swimming. It is worthwhile to highlight the valuable attempt to provide a humanistic explanation of why “Italians are the fastest 3000 m open-water master swimmers in the world”. Overall, authors manage to fulfill sufficiently their aim.

The authors could try to provide some further insights regarding the diffusion of masters sports in countries most successful in master open-water swimming. One further suggestion could be to try to find information regarding nutrition habits in master athletes in countries most successful in master open-water swimming.

Answer: We agree with the expert reviewer and insert a specific section to address these aspects: Apart from Italy, master swimming is also supported in other countries. In the United States of America, U.S. Masters Swimming promotes open water master swimming with providing details of clubs, events and training for future open-water master swimmers. In England, master swimmers can find a club and information about swimming technique and nutrition. Most probably, there are in other countries than Italy also very specific regions where open-water master swimmers are specifically supported. In Oregon, USA, Oregon Masters Swimming promotes lifelong fitness by offering organized swimming for adults of all ages. Also in Wales, master swimmers are supported specifically with information about swimming techniques and events. Eventually, these organizations also provide specific information about nutrition in open-water.

Reviewer 3 Report

Despite the present manuscript contains a huge sample of open water results along fifteen years, there are some serious gaps in the research design and the rationale of the study that make it unsuitable for publication (in the opinion of the present reviewer).  

First, the rationale for this study (athletes originating from a specific region or country can master specific sport disciplines) and the examples stated for this (east-african runners in long-distance running or Jamaican runners for sprints events) have very less to do with the present manuscript. This is mainly because the sample in the present research does not correspond to elite performers but to Master swimmers. But also, because the 3000m race event is not an official distance in both the World Swimming Championships or Olympic Games program.

Second, authors state in the title that “Italians are the fastest 3000 m open-water master swimmers in the world”. However, this statement is made upon the comparison of swimming times between fifteen different open-water events. Probably, external factors like water temperature, currents or circuit design were different and this undoubtedly affected the swimming times. It is well acknowledged that time or velocity comparison can be misleading in open water events due to the changing external conditions (water temperature, currents, circuit structure, …) between races (Baldassarre, 2017). In this way, it is not surprising that all the best times were obtained in the same race (page 3, line 117) “all the male and female ten best times in 3000 m open-water swimming occur in the same World Championship, which occurs in Riccione, Italy, 2012”.  

In relation to the data analysis, authors state that  “Italians are the fastest 3000 m open-water master swimmers in the world” but data shown in table 2 contradicts this statement.

Author Response

Reviewer 3

Despite the present manuscript contains a huge sample of open water results along fifteen years, there are some serious gaps in the research design and the rationale of the study that make it unsuitable for publication (in the opinion of the present reviewer).  

First, the rationale for this study (athletes originating from a specific region or country can master specific sport disciplines) and the examples stated for this (east-African runners in long-distance running or Jamaican runners for sprints events) have very less to do with the present manuscript.

Answer: Since there is very little research about the aspect of nationality in sports performance, we present the best-known results (running) for this topic. Then, we lead to our specific topic. There is only little research about nationality in swimming, and no research about master swimmers.

This is mainly because the sample in the present research does not correspond to elite performers but to Master swimmers. But also, because the 3000m race event is not an official distance in both the World Swimming Championships or Olympic Games program.

Answer: We agree with the expert reviewer. Research must no always investigate the most popular sports disciplines seen on TV, there are also other athletes who deserve respect for scientific investigations.

Second, authors state in the title that “Italians are the fastest 3000 m open-water master swimmers in the world”. However, this statement is made upon the comparison of swimming times between fifteen different open-water events. Probably, external factors like water temperature, currents or circuit design were different and this undoubtedly affected the swimming times. It is well acknowledged that time or velocity comparison can be misleading in open water events due to the changing external conditions (water temperature, currents, circuit structure, …) between races (Baldassarre, 2017). In this way, it is not surprising that all the best times were obtained in the same race (page 3, line 117) “all the male and female ten best times in 3000 m open-water swimming occur in the same World Championship, which occurs in Riccione, Italy, 2012”.  

Answer: We have analyzed ALL race results from ALL races for this specific sports discipline and found this specific result. Therefore, the results must be true. We expanded that specific section to: A potential explanation that Italian swimmers were the fastest in the same Championship (Riccione, 2012) that climatic and environmental conditions (e.g., water temperature) were quite favorable in that race. Open-water swimming races are influenced by extreme environmental conditions (e.g., water temperature, tides, currents, and waves) with an impact on performance, tactics and pacing. Time or velocity comparison can be misleading in open-water swimming events due to the changing external conditions (e.g., water temperature, currents, circuit structure, etc.) between races. Most probably external conditions were outstanding in that specific race and local athletes could profit. However, this does not explain why swimmers from other countries were not better than the Italian swimmers since environmental conditions were the same for all swimmers in that specific race.

In relation to the data analysis, authors state that  “Italians are the fastest 3000 m open-water master swimmers in the world” but data shown in table 2 contradicts this statement.

Answer: Please look at figures 1 and 2. It is stated in the results section ‘all the male and female ten best times in 3000 m open-water swimming occur in the same World Championship, which occurs in Riccione, Italy, 2012’. There are different approaches to define the ‘best’, e.g., the annual best, the annual best three, the 10 best ever, etc. Please see our approach in the method section.

Reviewer 4 Report

The study topic is interesting, but the manuscript requires major improvements. Also. the English should be carefully reviewed and improved.

All the manuscript needs to be written considering the journal instructions for authors and template.  Please review the authors, affiliations, and contacts according to the journal instructions for authors.

32 - “Germain” - It is “Germany”. Please review all countries throughout the text, some errors appear along the text (“Mexican” 125, for example).

“World Championships” it suggested to be abbreviated “WC”, too many times in the manuscript.

Sometimes “men” and “woman” in the text, “male” and “female” suggested.

Line 60 - “USA” - The first time United States of America appear in the document, it is suggested to be abbreviated, and since only “USA”.

61 “,” - space missing before “Australia”.

64 “It is interesting to note that there more literature is available about” - Please review the English.

70 - Too much space before “ultra”. The same in line 93 before “table”.

111 - It is suggested to indicate the total number for male, similarly to the case of female “The ten best times for women were from 7 different countries.”

118 - “occur” / 119 - “occurs” - Please review the English.

124 - Please standardize. In this paragraph and in other moments in the text, sometimes numbers used, other in not numbers but word.

Figure 1 - Type of letter and size should be in accordance with text of manuscript and journal instructions/template.

Figure 1 and 2 legends are not according to the first paragraph in results, please review.

138-139 - world championship in small caps, contrarily to many other occasions in the text. Please review.

Table 2 - Please confirm if the numbers form (type and size of letter) is in line with the text and Int. J. Environ. Res. Public Health guidelines

141 - 142 - “Akaike information criterion (AIC) finite sample corrected = 150249.218, and Omnibus test = p<0.001”. This procedure should be indicated and explained/described in methodology.

147 - Please confirm that the values in the order of the thousand must contain a comma or period. Sometimes in the manuscript in one form, sometimes in the other. Please standardize according to journal´s instructions.

166 - “Interesting complementary information was that countries that had never participated in other events participated when they were hosts (South Korea and Morocco)”. Please consider reformulate the English.

173 - Sometimes “age-groups”, others “age groups” - Please standardize. The same considering “open-water” and “open water”.

Please consider tables after text in results for a more comprehensive analysis for the readers of the manuscript. Too many tables in a row without introduction or text related to its content.

Table 4 - Legends should be placed with countries in full.

Table 4 – Please format the table content [e.g. (Number of participants (n= 9,247) = not aligned]. The same with “Mean Time (s)” (compare tables 3 and 4). Do not forget to confirm if number with comma or period.

178 - Table 5 - Legend should algo be placed describing the countries.

180 “m” space is required.

181 - “open 180 water championship”? Not understandable.

181 - Table 5 - Legend describing the countries. comma or period. Also race column should be aligned.

183 - Table 7 - Third line is not in accordance with other tables format.

188 - “also originate”. Please consider English reformulation.

196 - “and (vi)” not correct.

204 - “the most numerous starters”. Please consider English reformulation.

206-207. I believe it should be indicated that USA and Australia traditionally “dominate” swimming events, with a literature references.

224 – “that climatic and environmental conditions”. Please reconsider reformulate the English phrase construction.

230 - titles = small caps.

230-237- The web pages here indicated should be placed in “Methodology”, not in discussion.

251 - This paragraph should be developed, namely indicating benefits associated to physical and mental health with literature references.

256 - “in both terms of both” - Please review the English.

259 - “have local populaces” - Not understandable.

268 - Paragraph with limitations of the study (e.g. not all data available) and suggestions for further studies should be place before conclusions.

269 - Conclusions should be more developed and “future studies” suggestion should be placed previously to conclusions, as indicated above.

276 - Please correct, namely “The following statements should be used” / 279 - visualization, X.X.; supervision, X.X.; project.

281 - End point is missing. Also “” should be withdrawn.

Institutional Review Board Statement: “The study was conducted according to the guidelines of the Declaration of Helsinki.” – This is not indicated in the text and I have questions if suitable for this study, since it is not associated to ethical principles for medical research involving humans.

Conflicts of Interest: “The authors declare no conflict of interest.”. “” should be withdrawn.

Please carefully review the references format. For example, “9. SCOTT, R.A.; GEORGIADES, E.; WILSON, R.H.; GOODWIN, W.H.; WOLDE, B.; PITSILADIS, Y.P. Demographic”.

Author Response

Reviewer 4

The study topic is interesting, but the manuscript requires major improvements. Also. the English should be carefully reviewed and improved.

Answer: We hope we have addressed all specific comments of this reviewer.

 All the manuscript needs to be written considering the journal instructions for authors and template.  Please review the authors, affiliations, and contacts according to the journal instructions for authors.

Answer: We agree with the expert reviewer and checked again the whole manuscript to be consistent with the guidelines for authors.

 32 - “Germain” - It is “Germany”. Please review all countries throughout the text, some errors appear along the text (“Mexican” 125, for example).

Answer: We agree with the expert reviewer and corrected as requested.

 “World Championships” it suggested to be abbreviated “WC”, too many times in the manuscript.

Answer: We agree with the expert reviewer and corrected as requested.

 Sometimes “men” and “woman” in the text, “male” and “female” suggested.

Answer: We agree with the expert reviewer and corrected as requested.

 Line 60 - “USA” - The first time United States of America appear in the document, it is suggested to be abbreviated, and since only “USA”.

Answer: We agree with the expert reviewer and corrected as requested.

 61 “,” - space missing before “Australia”.

Answer: We agree with the expert reviewer and corrected as requested.

 64 “It is interesting to note that there more literature is available about” - Please review the English.

Answer: We agree with the expert reviewer and corrected as requested.

 70 - Too much space before “ultra”. The same in line 93 before “table”.

Answer: We agree with the expert reviewer and corrected as requested.

111 - It is suggested to indicate the total number for male, similarly to the case of female “The ten best times for women were from 7 different countries.”

Answer: We agree with the expert reviewer and we rewrote the excerpt and add the suggested information. (line 113)

 118 - “occur” / 119 - “occurs” - Please review the English.

Answer: We agree with the expert reviewer and changed to ‘Curiously, all the male and female ten best times in 3000 m open-water swimming were achieved in the same WC, which was held in Riccione, Italy, 2012’.

 124 - Please standardize. In this paragraph and in other moments in the text, sometimes numbers used, other in not numbers but word.

Answer: We agree with the expert reviewer and corrected as requested. Number up to ten are now in words.

 Figure 1 - Type of letter and size should be in accordance with text of manuscript and journal instructions/template.

Answer: We agree with the expert reviewer and corrected as requested.

 Figure 1 and 2 legends are not according to the first paragraph in results, please review.

Answer: We agree with the expert reviewer and we rewrote the excerpt that references the figures (lines 114/115, 118/119) and the legends of the figures (lines 133, 136).

138-139 - world championship in small caps, contrarily to many other occasions in the text. Please review.

Answer: As suggested earlier by this reviewer, we changed World Championship to WC

 Table 2 - Please confirm if the numbers form (type and size of letter) is in line with the text and Int. J. Environ. Res. Public Health guidelines

Answer: Table body is correct

 141 - 142 - “Akaike information criterion (AIC) finite sample corrected = 150249.218, and Omnibus test = p<0.001”. This procedure should be indicated and explained/described in methodology.

Answer: We agree with the expert reviewer and we rewrote the statistical analysis section by adding details about the procedure. (lines 107-110)

 147 - Please confirm that the values in the order of the thousand must contain a comma or period. Sometimes in the manuscript in one form, sometimes in the other. Please standardize according to journal´s instructions.

Answer: We changed throughout the whole manuscript

166 - “Interesting complementary information was that countries that had never participated in other events participated when they were hosts (South Korea and Morocco)”. Please consider reformulate the English.

Answer: We agree with the expert reviewer and changed to ‘An interesting complementary information was that countries that had never participated in other events participated when they were hosts (South Korea and Morocco)’.

 173 - Sometimes “age-groups”, others “age groups” - Please standardize. The same considering “open-water” and “open water”.

Answer: We agree with the expert reviewer and adapted throughout the whole manuscript

 Please consider tables after text in results for a more comprehensive analysis for the readers of the manuscript. Too many tables in a row without introduction or text related to its content.

Answer: We agree with the expert reviewer and changed as suggested

 Table 4 - Legends should be placed with countries in full.

Answer: We agree with the expert reviewer and changed as suggested

 Table 4 – Please format the table content [e.g. (Number of participants (n= 9,247) = not aligned]. The same with “Mean Time (s)” (compare tables 3 and 4). Do not forget to confirm if number with comma or period.

Answer: We aligned tables 3 and 4, reviewed the data and insert table legend (line 181)

 178 - Table 5 - Legend should algo be placed describing the countries.

Answer: We insert the legend as suggested (lines 184-186).

180 “m” space is required.

Answer: We agree with the expert reviewer and changed as suggested

 181 - “open 180 water championship”? Not understandable.

Answer: We agree with the expert reviewer and changed as suggested

 181 - Table 5 - Legend describing the countries. comma or period. Also race column should be aligned.

Answer: We insert the legend as suggested in table 6 (lines 189-191) and reviewed the data and formatting.

 183 - Table 7 - Third line is not in accordance with other tables format.

Answer: We format table 7 to maintain uniformity and insert table legend (lines 195-197).

 188 - “also originate”. Please consider English reformulation.

Answer: We agree with the expert reviewer and changed as suggested

 196 - “and (vi)” not correct.

Answer: We agree with the expert reviewer and changed as suggested

 204 - “the most numerous starters”. Please consider English reformulation.

Answer: We agree with the expert reviewer and changed as suggested

 206-207. I believe it should be indicated that USA and Australia traditionally “dominate” swimming events, with a literature references.

Answer: We agree with the expert reviewer and changed as suggested

 224 – “that climatic and environmental conditions”. Please reconsider reformulate the English phrase construction.

Answer: We agree with the expert reviewer and changed as suggested

 230 - titles = small caps.

Answer: We agree with the expert reviewer and changed as suggested

 230-237- The web pages here indicated should be placed in “Methodology”, not in discussion.

Answer: We agree with the expert reviewer and changed as suggested

 251 - This paragraph should be developed, namely indicating benefits associated to physical and mental health with literature references.

Answer: We agree with the expert reviewer and changed as suggested 

256 - “in both terms of both” - Please review the English.

Answer: We agree with the expert reviewer and changed as suggested

 259 - “have local populaces” - Not understandable.

Answer: We agree with the expert reviewer and changed as suggested

 268 - Paragraph with limitations of the study (e.g. not all data available) and suggestions for further studies should be place before conclusions.

Answer: We agree with the expert reviewer and changed as suggested

269 - Conclusions should be more developed and “future studies” suggestion should be placed previously to conclusions, as indicated above.

Answer: We agree with the expert reviewer and changed as suggested

 276 - Please correct, namely “The following statements should be used” / 279 - visualization, X.X.; supervision, X.X.; project.

Answer: We agree with the expert reviewer and changed as suggested

 281 - End point is missing. Also “” should be withdrawn.

Answer: We agree with the expert reviewer and changed as suggested

 Institutional Review Board Statement: “The study was conducted according to the guidelines of the Declaration of Helsinki.” – This is not indicated in the text and I have questions if suitable for this study, since it is not associated to ethical principles for medical research involving humans.

Answer: We agree with the expert reviewer and changed as suggested

 Conflicts of Interest: “The authors declare no conflict of interest.”. “” should be withdrawn.

Answer: We agree with the expert reviewer and changed as suggested

 Please carefully review the references format. For example, “9. SCOTT, R.A.; GEORGIADES, E.; WILSON, R.H.; GOODWIN, W.H.; WOLDE, B.; PITSILADIS, Y.P. Demographic”.

Answer: We agree with the expert reviewer and changed as suggested

Round 2

Reviewer 1 Report

Despite the authors improved the manuscript, I still feel the same as the first revision. As a coach, I see the manuscript to be interesting, presenting some descriptive data, and that should be interesting to present at a conference. There is novelty in the subject, considering that there is no previous study on this, but the results are merely descriptive. A deeper analysis and with some other outcomes should be interesting. I think the manuscript, in the way that is presented, do not meet the needed quality to be published in the current Journal.

Author Response

Reviewer 1

Despite the authors improved the manuscript, I still feel the same as the first revision. As a coach, I see the manuscript to be interesting, presenting some descriptive data, and that should be interesting to present at a conference. There is novelty in the subject, considering that there is no previous study on this, but the results are merely descriptive. A deeper analysis and with some other outcomes should be interesting. I think the manuscript, in the way that is presented, do not meet the needed quality to be published in the current Journal

Answer:  The aim of the present study was to verify where the athletes with the highest level of performance are from and also to identify which countries are most successful considering the number of participants. Knowing the countries with the highest level of performance allows studies to be done to understand how the characteristics of the athlete related to nationality can influence performance in the open water marathon events. Knowing the countries with the largest number of participants allows us to know which are the most successful countries in relation to the objective of maintaining the level of physical activity of master swimmers.

Reviewer 3 Report

Authors keep justifying the relevance of the present manuscript because “no research on this topic has performed before”. In the opinion of the present research this is not a valid argument for a research purpose.

But most importantly, the present reviewer has serious concerns on the interest of the analysis of performances by Master swimmers, which is a discipline developed by FINA (Federation Internationale Natation) to promote fitness among swimmers above 25 years old. When authors make parallelism between elite athletes and master athletes to justify the present research, they fail to build a valid rationale for the present research.

In addition, the authors select an open water event (3000m) which is not an official distance. This undoubtedly limits the potential and interest of the present study. Trying to draw conclusions from the nationality of swimmers in a non-official event can be misleading, as the homogeneous participation across different nationalities is non ensured (unlike official events like Olympic Games or World Championships).

Finally, authors insist on making velocity comparisons between open water events that have been performed with different external conditions. This is completely wrong as it has been repeatedly reported by previous research, as indicated in the previous reviewer report.

Author Response

Reviewer 3

Authors keep justifying the relevance of the present manuscript because “no research on this topic has performed before”. In the opinion of the present research this is not a valid argument for a research purpose.

Answer: We accept the personal opinion of this reviewer, and a supplement to the study's justification was included. The knowledge about the origin of the most successful athletes allows future studies to be developed in order to evaluate the specificities of these regions / peoples capable of generating differentiated results in aquatic marathon events.

But most importantly, the present reviewer has serious concerns on the interest of the analysis of performances by Master swimmers, which is a discipline developed by FINA (Federation Internationale Natation) to promote fitness among swimmers above 25 years old. When authors make parallelism between elite athletes and master athletes to justify the present research, they fail to build a valid rationale for the present research.

Answer: We accept the personal opinion of this reviewer, and a supplement to the study's justification was included. It is important to know which countries have the largest number of athletes participating in master swimming events, since these events were created to promote fitness among swimmers above 25 years old.

In addition, the authors select an open water event (3000m) which is not an official distance. This undoubtedly limits the potential and interest of the present study. Trying to draw conclusions from the nationality of swimmers in a non-official event can be misleading, as the homogeneous participation across different nationalities is non ensured (unlike official events like Olympic Games or World Championships).

Answer: We accept the personal opinion of this reviewer.

Finally, authors insist on making velocity comparisons between open water events that have been performed with different external conditions. This is completely wrong as it has been repeatedly reported by previous research, as indicated in the previous reviewer report.

Answer: We accept the personal opinion of this reviewer. We add at the end of the limitations ‘A further limitation is the fact that environmental conditions such as water temperature, tides, currents, and waves have not been considered. All these conditions might have an overall impact on performance, influencing tactics and pacing’.

Reviewer 4 Report

The manuscript significantly improved, congratulations, is close to the quality required for publication. Nevertheless, at this point requires some improvements (below indicated) and afterwards a careful and expected final analysis in attempt to find something to improve/correct. The English throughout the manuscript should be thoroughly reviewed and improved.

In affiliations, please consider include author´s abbreviation. Considering all the 10 authors, only 4 are mentioned.

96 - “A total of 9,687 results was analyzed”. Please carefully review the English here, and throughout the manuscript. In this specific case, “were analyzed”.

135 and 138 - “ten” instead of “10”. The same as in previous paragraphs. Standardization in important.

Figures 1 and 2 - The content letter format (font and size) is different compared to the manuscript text, please confirm and standardize.

Tables 3 and 4 - It is suggested table format aiming that these tables became similar to table 2. For example, column 3 (s) could be below the variable, as in the other variables. It also seems possible to reduce the space of all columns.

176 - “1128” and “6776”. I believe “1,128” and “6,776”, according to all manuscript criteria.

177 - Please standardize the size of the “-“. The same in 209-211 and 233-235.

202 - “ten”.

204 - “43 %” please change to “43%”.

205 - “68 %” please change to “68%”.

268 - “promotes open-water master swimming with providing” – I believe “with” in incorrect since it is not necessary in the phrase. Please confirm.

272-273 - “Most probably, there are in other countries than Italy also very specific regions where open-water master swimmers are specifically supported”. Please consider reformulate the English.

301 - “and were the most numerous starters during this period of 27 years and 15 editions all over the world”. Please consider reformulate the English.

305 - “K..”. One more point.

Please carefully double check the references in the final. One example, some end with end point, others not.

Author Response

Reviewer 4

The manuscript significantly improved, congratulations, is close to the quality required for publication. Nevertheless, at this point requires some improvements (below indicated) and afterwards a careful and expected final analysis in attempt to find something to improve/correct. The English throughout the manuscript should be thoroughly reviewed and improved.

Answer: We checked the manuscript again for potential errors

In affiliations, please consider include author´s abbreviation. Considering all the 10 authors, only 4 are mentioned.

Answer: We agree with the expert reviewer and changed as suggested

96 - “A total of 9,687 results was analyzed”. Please carefully review the English here, and throughout the manuscript. In this specific case, “were analyzed”.

Answer: We agree with the expert reviewer and changed as suggested

135 and 138 - “ten” instead of “10”. The same as in previous paragraphs. Standardization in important.

Answer: We agree with the expert reviewer and changed as suggested

Figures 1 and 2 - The content letter format (font and size) is different compared to the manuscript text, please confirm and standardize.

Answer: We agree with the expert reviewer and changed as suggested 

Tables 3 and 4 - It is suggested table format aiming that these tables became similar to table 2. For example, column 3 (s) could be below the variable, as in the other variables. It also seems possible to reduce the space of all columns.

Answer: We accept the opinion of this reviewer; however, we think that the change would difficult the table comprehension. We agree with the reviewer and reduce the space of all columns.

176 - “1128” and “6776”. I believe “1,128” and “6,776”, according to all manuscript criteria.

Answer: We agree with the expert reviewer and changed as suggested

177 - Please standardize the size of the “-“. The same in 209-211 and 233-235.

Answer: We agree with the expert reviewer and changed as suggested

202 - “ten”.

Answer: We agree with the expert reviewer and changed as suggested

204 - “43 %” please change to “43%”.

Answer: We agree with the expert reviewer and changed as suggested

205 - “68 %” please change to “68%”.

Answer: We agree with the expert reviewer and changed as suggested

268 - “promotes open-water master swimming with providing” – I believe “with” in incorrect since it is not necessary in the phrase. Please confirm.

Answer: We agree with the expert reviewer and changed as suggested

272-273 - “Most probably, there are in other countries than Italy also very specific regions where open-water master swimmers are specifically supported”. Please consider reformulate the English.

Answer: We agree with the expert reviewer and changed as suggested

301 - “and were the most numerous starters during this period of 27 years and 15 editions all over the world”. Please consider reformulate the English.

Answer: We agree with the expert reviewer and changed as suggested

305 - “K..”. One more point.

Answer: We agree with the expert reviewer and changed as suggested

Please carefully double check the references in the final. One example, some end with end point, others not.

Answer: We agree with the expert reviewer and changed as suggested